# Development of Therapy Based on the Exploration of Biological Events Underlying the Pathogenetic Mechanisms of Chronic Hepatitis B Infection

**DOI:** 10.3390/biomedicines11071944

**Published:** 2023-07-08

**Authors:** Sheikh Mohammad Fazle Akbar, Mamun Al Mahtab, Osamu Yoshida, Julio Aguilar, Guillen Nieto Gerardo, Yoichi Hiasa

**Affiliations:** 1Department of Gastroenterology and Metabology, Ehime University Graduate School of Medicine, Toon 791-0295, Japan; yoshidao.m.ehime@gmail.com (O.Y.); hiasa@m.ehime-u.ac.jp (Y.H.); 2Miyakawa Memorial Research Foundation, Tokyo 107-0062, Japan; 3Interventional Hepatology Division, Bangabandhu Sheikh Mujib Medical University, Dhaka 1000, Bangladesh; shwapnil@agni.com; 4Center for Genetic Engineering and Biotechnology, Havana 10400, Cuba; julio.aguilar@cigb.edu.com (J.A.); gerardo.guillen@cigb.edu.com (G.N.G.)

**Keywords:** cccDNA elimination, combination therapy, immune therapy, polyclonal immune modulator, vaccine therapy

## Abstract

According to the World Health Organization (WHO), an estimated 296 million people are chronically infected with hepatitis B virus (HBV). Approximately 15–25% of these people develop complications such as advanced chronic liver diseases (ACLDs). Mortality due to HBV-related complications accounted for an estimated 882,000 deaths in 2019. Potent preventive vaccines have already restricted new HBV infections, and several drugs are available to treat chronic HBV infections. However, the positive impacts of these drugs have been recorded in only a few patients with chronic HBV infection. These drugs do not show long-term efficacy and cannot halt the progression to complications. Thus, more effective and evidence-based therapeutic strategies need to be urgently developed for patients with chronic HBV infection. CHB is a pathological entity induced by HBV that progresses due to impaired host immunity. This indicates the inherent limitations of antiviral-drug-based monotherapy for treating patients with chronic HBV infection. Additionally, commercially available antiviral drugs are not available to patients in developing and resource-constrained countries, posing a challenge to achieving the following WHO goal: “Elimination of Hepatitis by 2030”. As such, this review aimed to provide insights regarding evidence-based and effective management strategies for chronic HBV infection.

## 1. Introduction

Developing a therapy for any pathology poses an important challenge to the contemporary disciplines of medical sciences. In the course of halting the progression of a pathogenesis, drugs may provide a cure or may have partial efficacy among patients. In the context of an infectious disease, a therapy is selected when the etiological agent of that disease is known. Based on the nature of the etiological agent and the pathogenic processes, complete cure, partial improvement, or containment of the etiological agent is expected. When this information is compiled with epidemiological data, a management strategy is developed. However, this process is not fixed for all infectious agents. For example, hepatitis B virus (HBV), a DNA virus with high tissue and species specificity, induces a range of pathological conditions ranging from asymptomatic HBV infection to advanced chronic liver diseases (ACLDs) [1,2,3,4]. HBV also causes different magnitudes of liver failure. HBV was discovered more than half a century ago, and a highly potent prophylactic vaccine was formulated within one decade of its discovery [5,6,7]. This preventive vaccine has been widely used, with an estimated global coverage of more than 80% to protect people from new HBV infection, with the help of the World Health Organization (WHO) and several other non-government organizations. However, drug coverage varies from country to country. 

In highly endemic areas, hepatitis B is most commonly spread from a mother to a child at birth (perinatal transmission) or through horizontal transmission (exposure to infected blood). This usually happens from an infected child to an uninfected child during the first five years of life. Hepatitis B is also spread by needle injuries, tattooing, piercing, and exposure to infected blood and body fluids, such as saliva and menstrual, vaginal, and seminal fluids [8,9]. The reuse of contaminated needles and syringes or sharp objects in either healthcare settings, the community, or among persons who inject drugs is also related to HBV transmission. Although sexual transmission is a possibility, it is more prevalent in unvaccinated persons with multiple sexual partners.

An estimated 296 million people across the world are chronically infected with HBV. Most of the people with chronic HBV infection express hepatitis B surface antigen (HBsAg) and HBV DNA in their sera [10]. Chronic HBV infection passes through four stages of pathogenesis [11]. Approximately 15–25% of those with chronic HBV infection are prone to developing ACLD [12,13,14]. Once CHB patients progress to ACLD, no curative treatment exists, and these patients are supported by symptomatic management. Patients with ACLD may develop ascites, esophageal hemorrhage, cerebral encephalopathy, and several other complications [15,16,17,18]. These symptoms can considerably compromise quality of life, posing extensive material and psychological costs to patients and nations. Usually, patients develop ACLD one or two decades after developing CHB. 

Several antiviral drugs have been developed during the last four decades to treat people with CHB. Two types of drugs have been extensively used to treat patients with CHB, including interferons (IFNs) and nucleoside analogs (NAs). IFNs are antiviral drugs with some immune modulatory potentials, whereas NAs are highly effective antiviral drugs for most of the CHB patients. The fundamental target of treating these patients with IFNs and NAs is to achieve HBsAg negativity as this is regarded as “functional care” [19,20,21,22]. Although these antiviral agents are capable of inducing “functional care” in some patients with CHB, this is not achieved in the majority of CHB patients [23,24,25,26,27,28,29,30,31,32]. 

These realities indicate that a new, innovative, and patient-friendly drug must urgently be developed for people with CHB to reduce HBV-related morbidity and mortality. However, commercially available antiviral medicines and several innovative drugs have not been developed owing to the fundamental factors of CHB pathogenesis. In this review, the limitation of commercially available antiviral drugs and their possible mechanisms are discussed. Next, a comprehensive description outlines the pros and cons of current innovative therapies for CHB. Finally, a blueprint of evidence-based therapy is provided to prevent HBV-related complications and to meet the WHO target of eliminating hepatitis by 2030 [33].

## 2. Commercially Available Antiviral Drugs and Their Scope and Limitation

Two groups of antiviral drugs are currently commercially available to treat those with CHB. These include (1) interferon drugs, which include IFNs and pegylated interferon (Peg-IFN). IFNs have been used to treat CHB since the early 1980s. With the advent of NAs, several NAs are now recommended for the treatment of CHB. Almost 25 years have passed since the first use of NAs for CHB patients. When these drugs (IFNs and NAs) were initially used to treat CHB, the target of therapy was to reduce HBV DNA in patients’ sera and achieve a seronegative status for hepatitis B e antigen (HBeAg) or seroconversion of HBeAg to anti-HBe. Notable professional international organizations, such as the American Association for the Study of Liver Diseases (AASLD), the European Association for the Study of the Liver (EASL), and the Asian Pacific Association for the Study of the Liver (APASL), provided IFN and NA treatment guidelines for patients with CHB [34,35,36]. 

IFNs represent a finite-duration mode of therapy, providing several advantages as a therapeutic option for treating CHB. They have antiviral properties, and many patients with CHB exhibited a loss of HBeAg and seroconversion to anti-HBeAg after receiving therapy with IFNs [37,38,39,40,41]. NAs can be orally administered and possess strong antiviral activity. The use of NAs results in HBV DNA reduction in almost all cases and HBV DNA negativity in the majority of CHB patients. The use of NAs also delays CHB progression to complications in some patients [42,43,44,45,46,47,48,49,50].

Along with these achievements, considerable limitations exist regarding the safety and efficacy of IFNs and NAs. IFNs are administered to patients via parenteral routes. Additionally, the use of IFNs entails considerable safety concerns; these drugs cannot be used in specific population groups. IFNs should not be used in those with decompensated liver disease, and this remains an absolute contraindication for Peg-IFN treatment because of the risk of IFN-induced hepatitis flares resulting in hepatic decompensation. Other important contraindications include severe psychiatric disorders (depression and suicidal ideation), severe cardiac disease, and autoimmune hepatitis (or other autoimmune disorders [51]. Moreover, IFNs are costly; even after achieving viral suppression in the sera and seroconversion to anti-HBe, the use of IFNs may not prevent progression to liver disease. Although this is a finite-duration treatment, the limitations of IFNs hinder their use for treating large populations of CHB patients in countries where CHB is endemic. 

Considerable optimism was generated after the introduction of NAs at the end of the 20th century due to several practical reasons. NAs can be orally administered, are comparatively cheaper, and safer NAs have been developed over time. NAs show strong antiviral properties. The use of NAs usually induces HBV DNA negativity in most patients with CHB. Entecavir (ETV), an oral nucleoside analog, is a potent inhibitor of HBV infection, with the effect being most evident in patients who regularly use ETV. If a patient adheres to treatment, virologic remission rates of >95% can be maintained with entecavir for 3–5 years. Tenofovir is also a potent antiviral drug for HBV and has an excellent safety profile. The limitation of NAs is related to their prolonged use and minimal effect on the covalently closed circular DNA (cccDNA) of HBV. Additionally, the immune modulatory capacities of NAs are limited. Taken together, an infinite-duration therapy such as treatment with an NA, if it has limited efficacy, is not a suitable therapeutic option for patients in developing and resource-constrained countries [52,53,54,55]. The limitations of commercially available antiviral drugs are summarized in Table 1.

Due to the properties of commercially available antiviral drugs and their inability to contain cccDNA, these drugs are unable to induce HBsAg negativity in those with CHB. These drugs are incapable of inducing requisite and protective immunity, especially cell-mediated immunity. However, these drugs are effective in containing the nucleic acid of HBV, but they do not play a role in protein synthesis in those with CHB. 

## 3. Genesis of CHB in HBV-Infected Patients 

HBV is a noncytopathic virus, and the virus is unable to directly damage hepatocytes. Thus, the role of HBV in inducing hepatic inflammation, damage of the liver, and progressive liver diseases is minimal or nonexistent. Additionally, HBV is not a direct inducer of fibrosis; thus, the progression to an ACLD is not due to the direct effect of HBV. Similarly, HBV is not directly carcinogenic, and HBV-related hepatocellular carcinoma (HCC) is not a direct result of the virus. Taken together, we are dealing with a noncytopathic virus that does not have cell destruction, fibrosis progression, and carcinogenic abilities. In this context, the question arises as to how ACLDs develop in patients infected with HBV. Several pieces of evidence and circumstantial findings indicate that HBV-related pathogenesis is induced and maintained by different mediators produced during HBV/host interactions [56,57,58,59,60,61,62,63]. In this context, we analyzed the role of the immune system in infectious pathologies.

Fundamentally, innate immunity should act as the first line of defense during HBV infection. The response to IFNs is inadequate in HBV infection [64]. Thus, a lower expression of IFN-signaling/stimulated genes and other factors, such as impaired activation of Toll-like receptor (TLR), may play a key role in establishing HBV infection [65]. 

Several researchers have documented the roles of different factors in various molecular and cellular events in the pathogenesis of HBV-related CHB. The liver is a lymphoid and classified organ that regulates several events in immunity. When HBV cannot be fully eradicated or its progression halted, the stimulation of hepatocytes and liver-resident cells of the immune system induces a large volume of proinflammatory immune modulators, including cytokines and chemokines. HBV is a stealth virus; after constant exposure to hepatitis, HBV-related antigens lead to immune exhaustion. When these factors act together, proinflammatory cytokines act as a trigger for liver damage. Maini et al. provided support for this concept in CHB patients, finding that proinflammatory non-antigen-specific cytokine levels are high in patients with CHB with increased HBV DNA and considerable liver damage. Additionally, HBV-antigen-specific immunity is upregulated in patients to control HBV DNA replication and liver damage [66,67,68]. Thus, the difference between pathogenic immunity and protective immunity is noticeable in the context of CHB. 

### 3.1. Progressive Hepatic Fibrosis in CHB Patients 

Advanced fibrosis in patients with CHB that ultimately progresses to advanced liver disease (ALD) is usually characterized by a progressive inflammatory hepatic microenvironment. *IL-1β* and *TNF-α* are upregulated in patients with advanced fibrosis, correlating with more severe inflammation [69]. The molecular mechanism of the fibrotic effect of *IL-1β* may be because of its capacity to elicit a fibrotic cascade in hepatic stellate cells (HSCs). Additionally, the role of *TGF-β1* is evident in CHB patients with advanced fibrosis, suggesting its critical function in activating myofibroblasts [70]. Taken together, these results show that the cross-talk between profibrotic and proinflammatory pathways contributes to the regulation of hepatitis fibrosis. 

### 3.2. Hepatocarcinogenesis in CHB Patients 

The mechanism leading to hepatocarcinogenesis in patients with CHB seems to be highly diverse, and this may be due to the integration of HBV DNA into the human genome along with other immunological factors. HCC usually develops after the development of ALD; however, in some patients, CHB may directly progress to HCC. Although integration is observed in billions of people with HBV infection, HCC only develops in a small fraction of these patients. In addition, the alteration in gene expression and the chromosomal instability due to previous hepatitis and liver cirrhosis may underlie this progression. Additionally, T-cell dysfunction, cytokine production, and inflammation-mediated alteration of specific signaling pathways play essential roles in the development of HCC, as seen in other carcinogenesis [71,72,73].

## 4. Developing Innovative Therapies for Patients with CHB

### 4.1. Concept of Innovative Drugs

Almost forty years have passed since the first use of antiviral drugs for treating people with CHB. Millions of people have been treated with these drugs globally. However, these drugs have mainly been used in advanced and rich countries, but most people who are chronically infected with HBV reside in developing and resource-constrained countries and are unaware of their HBV infection status. Providing treatment for the millions of patients with CHB in most developing and resource-constrained countries represents a formidable challenge to medical science. Drugs such as IFNs require parenteral administration, are costly, and have severe adverse effects; NAs require infinite-duration therapy and, thus, are unlikely to be suitable for accomplishing the international goals regarding HBV elimination. Several investigators have devoted considerable effort toward developing innovative drugs for the treatment of CHB. Studies aiming at developing alternative medicines for the treatment of CHB were initiated more than four decades ago. However, none of the drugs developed in these studies has received general acceptance, except for traditional IFNs and NAs. 

### 4.2. Major Types of Innovative Drugs

The innovative drugs used for the management of CHB may be divided into two main categories: The first-line drugs are those with antiviral properties. These drugs are assumed to be better than commercially available antiviral drugs such as IFNs and NAs (Table 2A). The next category represents drugs with immune modulatory capacities. These drugs can be further classified into two types: those aimed at inducing protective immunity (Table 2B) and those aimed at the restoration of impaired immunity in people with CHB (Table 2C). 

## 5. Development of New Antiviral Drugs: Clinical Implication for Management of CHB Patients 

In Table 2, we show how drug treatment for CHB may be improved via innovative antiviral drugs. We must assess if new antiviral drugs can resolve two critical issues facing CHB management: are they effective against cccDNA? Do they have a protective immune modulatory capacity? Entry inhibitors, a new type of antiviral drugs being developed, may block the entry of intrahepatic HBV; however, they would not be able to contain cccDNA. Clinical trials have identified the scope and limitations of these drugs, although they might play some role in containing cccDNA [74,75,76]. Similarly, antiviral drugs with RNA interference and CpAM properties would be able to contain cccDNA. Regarding the immune modulatory capacities of these drugs, no information is currently available. However, no indication exists that these drugs can induce protective immunity in CHB patients. Although some clinical trials are underway with entry inhibitors, other drugs have yet to be optimized for human use. The only drug that could alter CHB therapy would be a cccDNA inhibitor. These drugs have been analyzed in in vitro studies and animal models. If cccDNA can be completely controlled by antiviral drugs, this would be a breakthrough in the treatment of CHB. Once a drug with that capacity is produced, other factors could be tested in people with CHB to optimize such drugs [77,78,79,80]. In addition, checkpoint inhibitors, drugs regulating HBsAg production, engineered T cells, and Toll-like receptor agonists are new drugs in development [81].

## 6. Innovative Drugs Based on CHB Pathogenesis: Past Experiences and Present Realities Regarding Immune Therapy

Our primary aim in this review was to exhibit the path of drug development for those with CHB on the basis of the pathogenic mechanisms underlying the development of CHB. As HBV is a noncytopathic virus, liver damage, inflammatory changes, progressive fibrosis, and carcinogenesis are not directly attributable to HBV or its antigens and HBV-related antibodies. This has been further supported by critical analyses of the natural course of HBV infection. As of now, an estimated 2 billion people are infected with HBV; of these people, an estimated 296 million people, or 13%, develop chronic HBV infection and express HBV DNA and/or HBsAg in their sera. The management of these 296 million HBV-infected subjects represents a serious challenge for containing HBV infection. A critical aspect to achieving this aim is a deeper analysis of the natural course of disease progression among those with chronic HBV infection. Of the 296 million people with chronic HBV infection, approximately 20% would develop inflammation, and some of them would progress to develop ACLD, which causes about 882,000 deaths per year. People with chronic HBV infection who progress to ACLD, as well as those not progressing to HBV-related complications, harbor HBV DNA and/or HBsAg. These findings indicate that therapy should be targeted at a factor other than viral replication as HBV DNA is not a viable marker of liver damage in CHB patients. 

The focus of drug development should be shifted from viral control to liver damage. Circumstantial evidence has shown that both innate and adaptive immunities are distorted in CHB patients. Thus, immune therapy seems to be a method to develop innovative drugs for CHB treatment. In line with this, researchers have used cytokines, growth factors, immune modulators, and other immunogenic agents to upregulate the host immunity of patients with CHB. These have included cytokines such as interleukin (IL)-2, growth factors such as granulocyte-macrophage colony-stimulating factor, and different types of immune modulators [82,83,84,85,86,87,88,89,90]. The use of polyclonal immune modulators transiently received considerable optimism about their utility in treating CHB. However, none of these clinical trials exhibited sustained effects on CHB. These trials were conducted as pilot studies, so no long-term follow-up data are available, even though their authors commented about the favorable effects of these immune modulators in patients with CHB. Moreover, these studies did not explore the mechanism of action of these modulators. Notably, the role of polyclonal immune modulators could not be determined in CHB. However, properly designed phase I/II and III trials may identify some of the beneficial effects of polyclonal immune modulators in CHB treatment in the future. 

From a critical analysis, researchers have found that CHB patients are not immunocompromised. Instead, they have specific immune-response defects to the virus due to multiple causes. Additionally, the concepts of pathogenic immunity and protective immunity have surfaced regarding the treatment of CHB patients. However, antigen-nonspecific polyclonal immune modulators are ineffective for treating CHB. Additionally, safety concerns remain about the use polyclonal immune modulators. This may lead to the start of a new immune therapy regimen in patients with CHB using HBV-antigen-specific immune mediators.

Pol et al. first administered a HBsAg-based immune therapy in 1994 [91]. As HBsAg is an integral part of this preventive vaccine for HBV, this therapy was named “vaccine therapy”. After the initial excitement about vaccine therapies, several modifications of vaccine therapies were administered in people with CHB. The dosage, duration, and design of vaccine therapies using HBsAg were changed to improve outcomes. In addition, modified therapeutic vaccines were prepared via antigen/antibody complexation. Cell-based HBsAg vaccines were also used in patients with CHB, in which antigen-presenting dendritic cells were pulsed with HBsAg, and HBsAg-pulsed dendritic cells were used as a therapeutic vaccine [92,93,94,95,96,97,98,99,100,101,102,103]. Similar to vaccine therapies using HBsAg, complexed vaccines also initially created optimism, but they too failed to reach therapeutic application. Moreover, the nature of these studies and trials and the lack of prolonged follow-up prevented the understanding of the real strength of HBV-antigen-specific vaccine therapies. 

Hepatitis B core antigen (HBcAg)-based immune responses, especially HBcAg-based T cell responses, might be essential for the development of novel therapy for CHB [66,67,68]. This was evidenced by exploring the mechanism of viral persistency and liver damage in patients with CHB. Thus, an immune therapy regimen was developed during the last decade in which both HBsAg and HBcAg were used as a therapeutic vaccine in CHB patients. Clinical trials (phases I, II, and III) were accomplished with a HBsAg/HBcAg-based therapeutic vaccine for the treatment of naïve CHB in Bangladesh. This study showed that the HBsAg/HBcAg-based vaccine was extremely safe and moderately effective in controlling HBV replication and the progression of liver damage. Also, this therapeutic vaccine induced significantly higher frequencies of HBeAg negativity and seroconversion compared to other medications [104]. The antiviral and liver protecting effects of this therapeutic vaccine were sustained as data at the end of treatment (EOT) and at two, three, and five years after the EOT supported its effects [105,106,107,108]. The HBsAg/HBcAg-based vaccine was further developed by adding an adhesive agent and used in a clinical trial in Japan; a decrease in HBsAg was recorded in CHB patients [109]. Other researchers used HBsAg, HBcAg, and HBX antigen-based vaccines in patients with CHB. These vaccines were found be safe, but they could not attain their target in NA-treated patients [110]. 

The immune modulators listed in Table 2 [B] have been developed to restore host immunity. Some of these drugs have been tested in pilot studies or clinical trials; in other cases, they are being developed. The real potential of these drugs will be determining after data, especially long-term follow-up data, are available.

## 7. Role of Combination Therapy in Treating CHB

A role may exist for combination therapy using evolving and innovative antiviral drugs and immune modulators for managing CHB patients. Here, CHB pathogenesis indicates the involvement of the virus and host immunity. Although IFNs and NAs are potent antiviral drugs, they do not functionally cure chronic HBV infection. Additionally, the limitations of innovative therapies have been evidenced. To treat patients with CHB, the virus should be contained by antiviral drugs, and immune therapy should be used to induce protective immunity. Some trials have been conducted with antiviral drugs and immune modulators in people with CHB; however, the long-term follow-up data of these studies have not yet been published. Despite the situation in reality, combination therapy may be an appropriate therapy for CHB. 

## 8. Summary

The development of CHB therapy is urgently required globally. HBV is an intractable medical problem that, despite the emergence of a highly effective prophylactic vaccine, approximately one million patients die every year due to HBV-related pathologies. Additionally, all these deaths have been occurring despite the existence of two highly potent antiviral drugs on the market. From a public health viewpoint, an estimated 296 million people are living, permanent reservoirs of the virus. These people can spread HBV to uninfected individuals. HBV infection is usually a lifelong disease and induces suffering for both patients and nations. There are considerable concerns about the safety and efficacy of commercially available antiviral drugs for treating CHB patients and arresting progression to ACLD; there is a pressing need to develop evidence-based therapy for the management of these patients. As HBV is a noncytopathic virus, drugs targeting only HBV will not be efficacious. As CHB develops because of the complex impairment of host immunity, immune intervention may be one of the viable scientific approaches. Although immune intervention has been used for over forty years in CHB patients, the outcomes are still not optimistic. This, however, is not the fundamental limitation of the concept of immune therapy. After analyzing most of the immune therapeutic protocols, we found that most of the developed immune therapies were examined in pilot studies or small-scale clinical trials. Thus, long-term follow-up data are not available. The role of antigens in these immune therapies has not been optimized. Little effort has been devoted to optimizing the doses of immune modulators. Additionally, whether these agents are safe to use for a prolonged duration is unclear. The mechanisms underlying the positive impact of these immune therapies have not been elucidated. Only one immune therapy has published 5-year follow-up data. 

## 9. Conclusions

The development of therapy for CHB must consider two important factors: the viral aspect and the pathogenic mechanisms responsible for liver damage. Commercially available antiviral drugs are incapable of efficiently accomplishing both tasks. IFNs and NAs cannot eradicate cccDNA, and they provide little protective immunity to counter liver damage. Innovative antiviral drugs for CHB should be able to control cccDNA and be capable of restoring the immunity of those with CHB. Innovative immune therapeutic drugs should induce protective immunity and downregulate pathogenic immunity. Evolving antiviral drugs and immune therapies may directly or indirectly accomplish these goals, or the combined effects of both treatments may benefit patients with CHB. In conclusion, although several drugs are being developed as antiviral drugs to manage CHB, the virus should not be the target of drug development. Instead, the containment of cccDNA and the restoration of host immunity should be the focus. Additionally, a combination of innovative antiviral drugs may be able to contain cccDNA to a level similar to that of an asymptomatic and inactive HBV-infected person. 

## Figures and Tables

**Table 1 biomedicines-11-01944-t001:** Scope and limitations of commercially available antiviral drugs for management of patients with CHB.

Role in containing HBV DNA
2. Cannot eradicate cccDNA residing in the hepatocyte nucleus
3. Immune modulatory capacities of these drugs are minimal
4. Adequate cell-mediated immunity is not induced by these drugs

**Table 2 biomedicines-11-01944-t002:** Innovative antiviral drugs for containment of CHB.

**A.** Drugs targeting the virus (antiviral drugs) 1. Inhibition of entry of HBV (References [74,75,76]); 2. Inhibition of cccDNA in the nucleus of hepatocytes (References [77,78,79,80]); 3. Interfering with the activity of RNA (References [77,78,79,80]); 4. Core protein allosteric modulators (CpAMs) inducing incorrect virus assembly (References [77,78,79,80]).
**B.** Drug inducing protective immunity in CHB patients *1.* *Immune therapeutic agents to induce protective immunity in CHB patients (References [81,82,83,84,85,86,87,88,89,90].* *2.* *Immune therapy with HBV-antigen-specific immune modulators* a. HBsAg-specific immune therapy (References [91,92,93,94]); b. HBsAg plus anti-HBs complex (References [95,96,97]); c. HBsAg-based DNA vaccine (References [98,99,100]); d. HBV antigen-based cellular [101,102,103]);
**C.** Drugs to restore host immunity (Reference [81]) Checkpoint inhibitors; Drugs regulating HBsAg production; Engineered T cells; Toll-like receptor agonists.

## Data Availability

Not applicable.

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
