# Peer review of "Development of Therapy Based on the Exploration of Biological Events Underlying the Pathogenetic Mechanisms of Chronic Hepatitis B Infection"

_biomedicines, 2023, doi:10.3390/biomedicines11071944_

Round 1

Reviewer 1 Report (Previous Reviewer 1)

The reviewed version by Sheikh Mohammad Fazle Akbar et al. has improved the quality after following the reviewer’s recommendations. However, major changes are necessary to be accepted as a “Review” format in biomedicines. 

Major comments

·       Introduction. Authors should include the beneficial effect of achieving the “functional cure” of hepatitis B (HBsAg seroconversion). Moreover, the authors should introduce the limited efficacy of NA and IFN to achieve this functional cure.

·       Authors persist in using the acronym LC (liver cirrhosis) in lines 183 and 196

·       Despite the reviewer’s recommendation, authors persist in using extensive sentences that can be eliminated to reduce the extension of the sections and improve the clarity of their review

o   Section 1. Line 65 is not necessary “although many …..these patients”

o   Section 2. Line 125 is not necessarily “being most evident….use entecavir”. The last paragraph is redundant and not necessary. “Taken together, …..with CHB” 

o   Section 3. Lines 166-167 are not necessary “These events….HBV infection”

o   Section 4. Lines 211-212 are not necessary “These data are vital….for CHB”

o   Section 5. Lines 232-237 are not necessary “The treatment…CHB patients”

·       Section 6. Lines 312-327 can be summarized. Authors can include more information regarding the safety and efficacy of this type of therapy especially regarding virological and biochemical response, HBeAg, and HBsAg seroconversion.

·       Summary. Line 354 is not adequate. Therapy aims to avoid the progression to ACLD and lead to HBsAg seroconversion.

*Please, eliminate dramatic words such as “destroy” cccDNA in lines 140, 245, and 251,  “kill” hepatocytes in line 151, “destruction” of hepatitis in line 152, and “human” CHB patients in line 176.

*Please introduce abbreviations one time, for interferons (lines 73 and 87), nucleoside analogs (lines 73 and 89), and hepatitis B e antigen (lines 93 and 100).

*Please use the same abbreviation for pegylated interferon (lines 88 and 112).

Author Response

Dear Sir

Thanks for your mail. The queries have been seriously analyzed and corrections have been done. The altered portions have been shown by colorful shading in the manuscript.

Comments;

The reviewed version by Sheikh Mohammad Fazle Akbar et al. has improved the quality after following the reviewer’s recommendations. However, major changes are necessary to be accepted as a “Review” format in biomedicines.

Responses: Thanks for the understandings of the Reviewer.

Major comments

Introduction. Authors should include the beneficial effect of achieving the “functional cure” of hepatitis B (HBsAg seroconversion). Moreover, the authors should introduce the limited efficacy of NA and IFN to achieve this functional cure.

Authors persist in using the acronym LC (liver cirrhosis) in lines 183 and 196

Query

Despite the reviewer’s recommendation, authors persist in using extensive sentences that can be eliminated to reduce the extension of the sections and improve the clarity of their review

Response: I am extremely sorry for this mistake. These have been corrected.

Query

Section 1. Line 65 is not necessary “although many …..these patients”

Response: This has been deleted

Query

Section 2. Line 125 is not necessarily “being most evident….use entecavir”. The last paragraph is redundant and not necessary. “Taken together, …..with CHB” 

Response: All modifications suggested by the Reviewers have been accomplished.

Query

Section 3. Lines 166-167 are not necessary “These events….HBV infection”

Response: These have been deleted.

Query

Section 4. Lines 211-212 are not necessary “These data are vital….for CHB”

Response: These have been deleted.

Query

Section 5. Lines 232-237 are not necessary “The treatment…CHB patients”

Response: These have been deleted.

  •  

Section 6. Lines 312-327 can be summarized. Authors can include more information regarding the safety and efficacy of this type of therapy especially regarding virological and biochemical response, HBeAg, and HBsAg seroconversion.

Response: This has been summarized (Line 302 to 313)

Query

Summary. Line 354 is not adequate. Therapy aims to avoid the progression to ACLD and lead to HBsAg seroconversion.

Response: This has been done

Comments on the Quality of English Language

Query

*Please, eliminate dramatic words such as “destroy” cccDNA in lines 140, 245, and 251,  “kill” hepatocytes in line 151, “destruction” of hepatitis in line 152, and “human” CHB patients in line 176.

Response: This has been done.

Query

*Please introduce abbreviations one time, for interferons (lines 73 and 87), nucleoside analogs (lines 73 and 89), and hepatitis B e antigen (lines 93 and 100).

*Please use the same abbreviation for pegylated interferon (lines 88 and 112).

Response: This has been done

Reviewer 2 Report (Previous Reviewer 2)

At the very least, thoses "TABLES",I am not sure if they can be categorized into tables, should be presented with related citations and readable informations. 

None.

Author Response

Sir

Thanks for your mail and understanding the compilation of the article. The araticle has been revised as per your recommendation and the altered areas have been marked by colored shadings.

Query

At the very least, thoses "TABLES",I am not sure if they can be categorized into tables, should be presented with related citations and readable informations.

Responses

As per your recommendation, the Table has been shown in tabulated format.

Also, all relevant references have been given in the Table.

Round 2

Reviewer 1 Report (Previous Reviewer 1)

Comments to Authors:

The reviewed version by Sheikh Mohammad Fazle Akbar et al. has improved but minor changes are necessary to be accepted as a “Review” format in biomedicines.

Comments

·       Title. Authors should include the article “the” before “exploration” 

·       Despite the reviewer’s recommendation, authors persist in using extensive sentences that can be eliminated to improve the clarity of their review. Section 2. Line 141-147 is redundant and it is not necessary “Taken together, …..with CHB” 

·       Section number 5. Line 226. Please, eliminate the number 5 from the beginning of the paragraph and correct its position.

·       Despite the reviewer’s recommendation, authors persist in using repeated words and abbreviations such as nucleoside analogs (NAs)(lines 72 and 91), using words without the abbreviation such as Entecavir (ETV)(lines 124 and 126), and using abbreviations without the first explanation such as cccDNA (line 128), HCC (line 153), and ALD (line 179)

Despite the reviewer’s recommendation, authors persist in using dramatic words such as “destroy” cccDNA in line 232“, “destruction” of cccDNA in line 364

Author Response

Please provide a point-by-point response to the reviewer’s comments and either enter it in the box below or upload it as a Word/PDF file. Please write down "Please see the attachment." in the box if you only upload an attachment. An example can be found here (/bundles/mdpisusy/attachments/Author/Example for author to respond reviewer - MDPI.docx?1e53c61ae425d6e8).

Comments to Authors: The reviewed version by Sheikh Mohammad Fazle Akbar et al. has improved but minor changes are necessary to be accepted as a “Review” format in biomedicines.

Response:

Comments

Title. Authors should include the article “the” before “exploration” ·

Response: As per the recommendation of the Reviewer, “the” is included before “exploration”

Query

Despite the reviewer’s recommendation, authors persist in using extensive sentences that can be eliminated to improve the clarity of their review. Section 2. Line 141-147 is redundant and it is not necessary “Taken together, …..with CHB” ·

Response: The sentences as mentioned by the Reviewer have been deleted.

Query

Section number 5. Line 226. Please, eliminate the number 5 from the beginning of the paragraph and correct its position. ·

Response: As per the recommendation of the Reviewer, this has been done.

Despite the reviewer’s recommendation, authors persist in using repeated words and abbreviations such as nucleoside analogs (NAs)(lines 72 and 91), using words without the abbreviation such as Entecavir (ETV)(lines 124 and 126), and using abbreviations without the first explanation such as cccDNA (line 128), HCC (line 153), and ALD (line 179)

Response: These have been corrected.

Query

Despite the reviewer’s recommendation, authors persist in using dramatic words such as “destroy” cccDNA in line 232“, “destruction” of cccDNA in line 364

Response: These have been corrected.

Reviewer 2 Report (Previous Reviewer 2)

None.

None.

Author Response

Response to Reviewer 2

Comments and Suggestions for Authors

None

Response: Thanks for your understanding.

Comments on the Quality of English Language: None

Response: Thanks for your understanding.

This manuscript is a resubmission of an earlier submission. The following is a list of the peer review reports and author responses from that submission.

Round 1

Reviewer 1 Report

The study by Sheikh Mohammad Fazle Akbar et al. is a narrative review regarding the properties and limitations of current antiviral treatments to achieve chronic hepatitis B (CHB) elimination. 

However, authors should make important changes to improve their narrative review’s quality and be accepted in biomedicines. 

1.     Abstract.

a.     Please, include the aim of this review in the abstract.

2.     Introduction.

a.     The extension of the introduction is excessive. Please review and summarize this section.

b.     Please use Advanced Chronic Liver Disease (ACLD) better than liver cirrhosis (LC)

c.     Line 64, Please,  include more accurate information about current antiviral treatments.

d.     Please avoid repetitions. i.e the sentence “….Elimination of Hepatitis by 2030 a target of the WHO…” is repeated in lines 20-21, 77-78, 103-104, 221-222, 290-291, and 403-404

3.     Available treatments

a.     Please, provide more accurate information about the efficacy and safety of the current treatments.

b.     Line 125. Please,  include more details about “specific groups”

c.     Please, avoid personal opinions ie line 135 “The pessimistic parts of NAs…”

4.     Table 1. Please avoid questions in the title “What commercially…or what are...” and avoid dramatic words such as “destroy cccDNA” or “Able to destroy…” 

5.     Chronic hepatitis B section

a.     Please, avoid subjective comments as in Line 155. “bird’s eye view…” or “…is insignificant, even if any” or in Line 222 “This target will never be achieved..” 

b.     Please simplify long sentences as lines 164-166 “The exact and complete mechanism underlying these queries may not be provided based on our contemporary insights about HBV pathogenesis and the development of CHB, LC, and HCC” and ”… maintained by, if not only, but mainly by, different mediators…” or “…these events are not crucial for this communication as we would mainly discuss the pathogenesis of CHB, LC, and HCC, not the mechanism underlying the establishment of HBV infection following being infected…”

6.     Tables 2, 3A, and 3B can be shown in a combined table

7.     Development of new drugs. 

a.     Please avoid subjective expressions such as “Treatment with antiviral drugs for CHB patients is comparable to "Old wine in a new bottle." 

b.     Please, do not repeat information about current NA/IFNs in this section. Lines 250-257 are not necessary.

c.     Please include conclusions (lines 271-273) in the “conclusions sections”. 

8.     Immune therapy

a.     Please do not repeat the information. Lines 283-285 are explained in the “Genesis of CHB” section

b.     Please avoid information without bibliographic references, ie lines 308-313, lines 315-321, and lines 326-328

c.     Please, develop the last paragraph (lines 355-358) in a new section with more detailed information and references.

9.     Summary

a.     Please, develop the last paragraph (lines 383-386) in a new section regarding combined therapy and included the important references

10.  Conclusions

a.     Please, shorten this section. Lines 394-404 are repetitive

The current version’s lack of brevity and clarity is a major problem. 

Author Response

Query:

The study by Sheikh Mohammad Fazle Akbar et al. is a narrative review regarding the properties and limitations of current antiviral treatments to achieve chronic hepatitis B (CHB) elimination. 

Response:

Thank you very much for you your understandings

However, authors should make important changes to improve their narrative review’s quality and be accepted in biomedicines

  1. Abstract.
  2. Please, include the aim of this review in the abstract.

Response: The aim of the study has been given in the Abstract (Line 28-30).

  1. Introduction.
  2. The extension of the introduction is excessive. Please review and summarize this section.

Response: I am sorry for the extended Introduction. The major points relevant to the scientific merit of the article have been comprehensively presented in revised version of the manuscript. The most relevant information has been concisely presented in INTRODUCTION. Additionally, both authors suggested to provide some more information about some specific topic. Thus, although the size of the INTRODUCTION has not been reduced significantly, the usage of repeating information has been avoided. In addition, some new information has been added (Revised Introduction, Line 34-92).

  1. Please use Advanced Chronic Liver Disease (ACLD) better than liver cirrhosis (LC)

Response: This has been done (Line 68).

  1. Line 64, Please, include more accurate information about current antiviral treatments.

Response: This comment of the Reviewer has been addressed by providing a balanced information of the commercially available drugs for treating CHB. These can be found in subchapter 2 (INTRODUCTION and Chapter 3).

  1. Please avoid repetitions. i.e the sentence “…. Elimination of Hepatitis by 2030 a target of the WHO…” is repeated in lines 20-21, 77-78, 103-104, 221-222, 290-291, and 403-404

Response: I am sorry for repetition. Repeated expressions have been deleted..

  1. Available treatments
  2. Please, provide more accurate information about the efficacy and safety of the current treatments.

Response: These have been accomplished.

  1. Line 125. Please, include more details about “specific groups”

Response: These have been discussed (Line 120-125)

  1. Please, avoid personal opinions ie line 135 “The pessimistic parts of NAs…”

Response: These have been done.

  1. Table 1. Please avoid questions in the title “What commercially…or what are...” and avoid dramatic words such as “destroy cccDNA” or “Able to destroy…”

Response: These have been done.

  1. Chronic hepatitis B section
  2. Please, avoid subjective comments as in Line 155. “bird’s eye view…” or “…is insignificant, even if any” or in Line 222 “This target will never be achieved..” 

Response: These have been done.

  1. Please simplify long sentences as lines 164-166 “The exact and complete mechanism underlying these queries may not be provided based on our contemporary insights about HBV pathogenesis and the development of CHB, LC, and HCC” and ”… maintained by, if not only, but mainly by, different mediators…” or “…these events are not crucial for this communication as we would mainly discuss the pathogenesis of CHB, LC, and HCC, not the mechanism underlying the establishment of HBV infection following being infected…”

Response: These have been accomplished.

  1. Tables 2, 3A, and 3B can be shown in a combined table

Response: This has been compiled to a new Table (Table 2).

  1. Development of new drugs. 
  2. Please avoid subjective expressions such as “Treatment with antiviral drugs for CHB patients is comparable to "Old wine in a new bottle." 

Response: These have been deleted.

  1. Please, do not repeat information about current NA/IFNs in this section. Lines 250-257

are not necessary.

Response: These have been deleted.

  1. Please include conclusions (lines 271-273) in the “conclusions sections”. 

Response: OK. This has been done

  1. Immune therapy
  2. Please do not repeat the information. Lines 283-285 are explained in the “Genesis of CHB” section

Response: These have been deleted

  1. Please avoid information without bibliographic references, ie lines 308-313, lines 315-321, and lines 326-328

Response: This has been addressed.  

  1. Please, develop the last paragraph (lines 355-358) in a new section with more detailed information and references.

Response: The contents of these lines have been discussed. .

  1. Summary
  2. Please, develop the last paragraph (lines 383-386) in a new section regarding combined therapy and included the important references

Response: This has been discussed in new chapter (Chapter 7).

  1. Conclusions
  2. Please, shorten this section. Lines 394-404 are repetitive

Response: This has been shortened.

Reviewer 2 Report

Sheikh Mohammad Fazle Akbar et al conducted a review on the development of CHB treatments, which remains an urgent unmet medical need for the eradication of HBV. The manuscript appears to have been written haphazardly, containing numerous rudimentary facts about HBV, CHB, and its treatments that are misleadingly presented, thereby raising significant doubts about the authors' academic rigor. Several paragraphs are challenging to comprehend, and the manuscript may require editing by a native speaker to improve its overall clarity and coherence. Moreover, the pathogenesis of HBV and its treatment are discussed separately, rather than being thoroughly integrated, highlighting the need for substantial revisions to the manuscript's structure.Additional comments are listed below:

Major issue:

1.        In lines 17-23 and 145, the authors suggest that marketed drugs have potent antiviral capabilities but do not contribute to the elimination of HBV. This statement requires careful rephrasing, as numerous clinical trials have demonstrated that Nucleos(t)ide analogues (NAs) and Interferon-alpha (IFN-a) based therapies do promote HBV eradication in a small proportion of patients.

2.        Line 80 states that chronic hepatitis B (CHB) is synonymous with liver inflammation, which is incorrect. In reality, there are four stages of chronic HBV infection, and inflammation does not occur during the immune tolerance phase.

3.        In the introduction, the authors fail to acknowledge that IFN-a based therapies are also marketed drugs for the treatment of HBV. In fact, IFN-a administration can lead to the functional cure of CHB, albeit at a low frequency. It remains unclear whether the authors deliberately or carelessly omitted this fact.

4.        All tables presented in this manuscript are too concise for a comprehensive review. The authors should reformat them to exhibit more meaningful information, such as categories, medications, targets, clinical trials, advantages, disadvantages, and references.

5.        The summary and conclusion sections are inadequate for a comprehensive review. These sections should be expanded to provide a more comprehensive overview of the manuscript's key findings and their implications.

Minor issue:

1.        Line 45 states that the discovery of the Australia antigen was "nearly half a century ago." However, it was actually more than half a century ago, in 1965. Additionally, the original articles describing these discoveries should be cited in the references.

2.        In line 46, the phrase "in almost all countries" could be replaced with "an estimated global coverage of 83%."

3.        Line 51 should have the word "permanent" removed.

4.        In line 52, the transmission routes of HBV should be listed instead of just saying "various means."

5.        Line 52 should have the phrase "out of the two billion HBV-infected person" removed.

6.        In line 53, the phrase "a considerable number" should be specified as an exact frequency.

7.        The two abbreviations used in line 56 are incorrect.

8.        Line 59 could be rephrased to say "established" instead of "triggered."

9.        In line 65, it should be noted that many medications and surgeries have been developed to treat liver cirrhosis (LC) and hepatocellular carcinoma (HCC).

10.     Line 74 should be changed to say "to develop" instead of "to developing strategies."

11.     Line 81 should use the phrase "limited efficacy" instead of "minimally active."

12.     In line 86, it should be noted that rcDNA is integrated, not cccDNA.

13.     Line 108 could be rephrased to say "approved medications for the treatment of CHB."

14.     In line 155, the phrase "bird's eye view" could be changed to "aerial view," still, it may be not appropriate in this context.

Several paragraphs are challenging to comprehend, and the manuscript may require editing by a native speaker to improve its overall clarity and coherence. 

Author Response

Reviewer 2

Comments and Suggestions for Authors

Sheikh Mohammad Fazle Akbar et al conducted a review on the development of CHB treatments, which remains an urgent unmet medical need for the eradication of HBV. The manuscript appears to have been written haphazardly, containing numerous rudimentary facts about HBV, CHB, and its treatments that are misleadingly presented, thereby raising significant doubts about the authors' academic rigor. Several paragraphs are challenging to comprehend, and the manuscript may require editing by a native speaker to improve its overall clarity and coherence. Moreover, the pathogenesis of HBV and its treatment are discussed separately, rather than being thoroughly integrated, highlighting the need for substantial revisions to the manuscript's structure. Additional comments are listed below:

Response: On the basis of the comments of the Reviewer, the manuscript has been revised. The comments of the Reviewers seem to be relevant in the context of the complexity of the HBV pathogenesis. These facts have been addressed by the authors in the revised manuscript. Nevertheless, we appreciate the comments of the Reviewers and we assume that these constructive comments have positively contributed to improve the article.

Major issue:

  1. In lines 17-23 and 145, the authors suggest that marketed drugs have potent antiviral capabilities but do not contribute to the elimination of HBV. This statement requires careful rephrasing, as numerous clinical trials have demonstrated that Nucleos(t)ide analogues (NAs) and Interferon-alpha (IFN-a) based therapies do promote HBV eradication in a small proportion of patients.

Response: These expressions have been altered as per recommendation of the Reviewer in the revised manuscript. Also, descriptions have been provided to highlight the role of IFNs and NA in treating CHB (Line 107-116). Also, this has been shown in the entire manuscript.

  1. Line 80 states that chronic hepatitis B (CHB) is synonymous with liver inflammation, which is incorrect. In reality, there are four stages of chronic HBV infection, and inflammation does not occur during the immune tolerance phase.

Response: The expression of the reviewer is correct. We have also modified this expression in revised manuscript.

  1. In the introduction, the authors fail to acknowledge that IFN-a based therapies are also marketed drugs for the treatment of HBV. In fact, IFN-a administration can lead to the functional cure of CHB, albeit at a low frequency. It remains unclear whether the authors deliberately or carelessly omitted this fact.

Response: The role of interferon has been acknowledged and discussed in the INTRODUCTION (Line 35-92) .

  1. All tables presented in this manuscript are too concise for a comprehensive review. The authors should reformat them to exhibit more meaningful information, such as categories, medications, targets, clinical trials, advantages, disadvantages, and references.

Response: The Tables have been reformatted. Tables 2 and 3A and B have reconstituted in Table 2.

  1. The summary and conclusion sections are inadequate for a comprehensive review. These sections should be expanded to provide a more comprehensive overview of the manuscript's key findings and their implications.

Response: These have been accomplished according to the comments of the Reviewer (Line 362-401.

Minor issue:

  1. Line 45 states that the discovery of the Australia antigen was "nearly half a century ago." However, it was actually more than half a century ago, in 1965. Additionally, the original articles describing these discoveries should be cited in the references.

Response: This has been corrected.

  1. In line 46, the phrase "in almost all countries" could be replaced with "an estimated global coverage of 83%."

Response: This has been corrected.

  1. Line 51 should have the word "permanent" removed.

Response: This has been removed.

  1. In line 52, the transmission routes of HBV should be listed instead of just saying "various means."

Response: This has been done.

  1. Line 52 should have the phrase "out of the two billion HBV-infected person" removed.

Response: This has been omitted

  1. In line 53, the phrase "a considerable number" should be specified as an exact frequency.

Response” This has been given.

  1. The two abbreviations used in line 56 are incorrect.

Response: Corrected.

  1. Line 59 could be rephrased to say "established" instead of "triggered."

Response; This has been done.

  1. In line 65, it should be noted that many medications and surgeries have been developed to treat liver cirrhosis (LC) and hepatocellular carcinoma (HCC).

Response: These have been done.

  1. Line 74 should be changed to say "to develop" instead of "to developing strategies."

Response: These have been done.

  1. Line 81 should use the phrase "limited efficacy" instead of "minimally active."

Response: These have been done.

  1. In line 86, it should be noted that rcDNA is integrated, not cccDNA.

Response: These have been done.

  1. Line 108 could be rephrased to say "approved medications for the treatment of CHB."

Response: These have been done.

  1. In line 155, the phrase "bird's eye view" could be changed to "aerial view," still, it may be not appropriate in this context

Response: These have been done.

Round 2

Reviewer 1 Report

Major comments

·       The introduction persists being very extensive. 

·       Authors persist in using the acronym liver cirrhosis (LC) which is not accepted, despite the recommendation to use Advanced Chronic Liver Disease (ACLD) in lines 16, 48, 70, 72, 74, 78, 160, 164, 167, 177, 194, 206, 210, 282, 283 and 373.

·       The authors have not included accurate information (efficacy and safety) about the current NAs (tenofovir, entecavir, and TAF)

·       The authors persist to repeat the same sentences such as “….Elimination of Hepatitis by 2030 a target of the WHO”  in lines 28, 82, 92, 223, and 279 despite the reviewer’s recommendation.

·       The authors have not eliminated dramatic words such as “Able to destroy…” in line 1 of Table 1

·       Despite the reviewer’s recommendation, authors persist in using the same extensive sentences such as “The exact and complete mechanism underlying these queries may not be provided based on our contemporary insights about HBV pathogenesis and the development of CHB, LC, and HCC” in lines 165-167 and ”… maintained by, if not only, but mainly by, different mediators…” in line 169 or “…these events are not crucial for this communication as we would mainly discuss the pathogenesis of CHB, LC, and HCC, not the mechanism underlying the establishment of HBV infection following being infected…” in lines 176-178

·       The included information in the new section regarding combined therapy is unprecise and very limited 

The reviewed version by Sheikh Mohammad Fazle Akbar et al. persists with important problems of clarity, brevity, and precision despite the reviewer’s recommendations

Author Response

Response to Reviewer 1

Thanks, the honorable Reviewer for several constructive comments these have been addressed as per the recommendations of the Reviewer. Also, changes have been shown by Yellow shading. The article has also been checked by MDPI English checking services.

Major comments · The introduction persists being very extensive. · Authors persist in using the acronym liver cirrhosis (LC) which is not accepted, despite the recommendation to use Advanced Chronic Liver Disease (ACLD) in lines 16, 48, 70, 72, 74, 78, 160, 164, 167, 177, 194, 206, 210, 282, 283 and 373.

Response: The Introduction has been truncated. The expressions have been changed according to the suggestions of the Reviewer.

The authors have not included accurate information (efficacy and safety) about the current NAs (tenofovir, entecavir, and TAF) · T

Response: This has been incorporated.

The authors persist to repeat the same sentences such as “….Elimination of Hepatitis by 2030 a target of the WHO” in lines 28, 82, 92, 223, and 279 despite the reviewer’s recommendation. ·

Response: These expressions have been omitted.

The authors have not eliminated dramatic words such as “Able to destroy…” in line 1 of Table 1 · Despite the reviewer’s recommendation, authors persist in using the same extensive sentences such as “The exact and complete mechanism underlying these queries may not be provided based on our contemporary insights about HBV pathogenesis and the development of CHB, LC, and HCC” in lines 165-167 and ”… maintained by, if not only, but mainly by, different mediators…” in line 169 or “…these events are not crucial for this communication as we would mainly discuss the pathogenesis of CHB, LC, and HCC, not the mechanism underlying the establishment of HBV infection following being infected…” in lines 176-178

Response: All recommendations and suggestions of the Reviewer have been accepted and reflected in a revised version of the manuscript.

Comments on the Quality of English Language

The reviewed version by Sheikh Mohammad Fazle Akbar et al. persists with important problems of clarity, brevity, and precision despite the reviewer’s recommendations

Response: The article has been checked for English usage by the MDPI English checking system

Reviewer 2 Report

While I appreciate the effort to address some of my concerns, I still find the tables presented in the revised manuscript insufficient to meet the acceptance criteria.

The manuscript requires editing by a native speaker to improve its overall clarity and coherence.

Author Response

Reviewer 2

Comments and Suggestions for Authors

While I appreciate the effort to address some of my concerns, I still find the tables presented in the revised manuscript insufficient to meet the acceptance criteria.

Response: Thank you very much for your understandings. The Tables have been revised. However, if there is any suggestion, we would do that.

Comments on the Quality of English Language The manuscript requires editing by a native speaker to improve its overall clarity and coherence.

Response: The manuscript has been checked by the MDPI English checking system
